# Land Surface Temperature and Urban Policies: The Ferrara City Case Study

Gianni Di Pietro, Emilio Marziali, Cristina Montaldi and Francesco Zullo *

Department of Civil, Construction-Architectural and Environmental Engineering, University of L'Aquila, Piazzale E. Pontieri, 1, Monteluco di Roio, 67100 L'Aquila, Italy; gianni.dipietro@graduate.univaq.it (G.D.P.); emilio.marziali@graduate.univaq.it (E.M.); cristina.montaldi@graduate.univaq.it (C.M.)
* Correspondence: francesco.zullo@univaq.it; Tel.: +39-0862434104

**Abstract:** Today's global challenges are increasingly complex, and forecast scenarios show a general increase in risks that could compromise human permanence in some areas of the planet. In this context, cities have a key role, both because they concentrate an increasing number of inhabitants and because they will be among the first areas to feel these effects. As pointed out by the IPCC, addressing these challenges requires a redefinition of the organization of urban spaces by assigning, more or less explicitly, a key role to spatial planning. Urban and territorial planning may be the main tool in the regulation of transformation processes. Planning has a crucial role, especially if territorial transformations are no longer mainly linked to expansive logics. In this case, it is possible to orient urban choices and policies towards a sustainable use of resources, including land resources that continue to be overexploited. Starting from these assumptions, the present work intends to analyze the relationship between the LST (Land Surface Temperature) extracted from the data provided by MODIS (Moderate-resolution Imaging Spectroradiometer) and the level of soil sealing within the municipality of Ferrara in northern Italy. The reference period is between 2015 and 2021. The objective is dual. The first is investigating how the environmental matrix can influence the temperature values detected; the second is investigating how the implementation of transformative forecasts provided by the urban planning tool in force, could influence the thermal comfort of the study area.

**Keywords:** Land Surface Temperature (LST); soil sealing; urban planning; territorial planning; urban forecast scenario

## 1. Introduction

The challenges facing cities in the 21st century, especially the larger ones, are many, from population growth in urban areas to the increased presence of the elderly among its inhabitants, and from environmental degradation to the scarcity of available resources [1–4].

Urban sprawl is one of the most relevant issues in this regard, as it plays a crucial role in economic development and social welfare; nevertheless, concerns about its detrimental effects on the environment, both locally and globally, are continuously increasing. The process of urban sprawl is one of the most visible, irreversible, and rapid types of land cover/land use change in contemporary human history, and is a key factor in many large-scale environmental and social changes. These changes often result in high temperatures, changes in precipitation, and increased concentrations of air pollution, as impervious surfaces can alter surface albedo and evapotranspiration while also releasing aerosols and anthropogenic heat [5,6].

In this context, the IPCC (Intergovernmental Panel on Climate Change) has called for a redefinition of urban and territorial organization to address the challenges [7]. Specifically, it identifies four categories of key risks for Europe: risks of heat waves on populations and ecosystems, risks to agricultural production, risks of water scarcity, and risks produced by increased frequency and intensity of floods [8–11]. The intensity with which each risk

manifests itself increases with increasing global warming [12]. Among the four categories reported, heat waves on populations and ecosystems today represent an increasingly tangible risk whose effects are reflected in daily life [13,14]. According to the IPCC, the number of deaths and people at risk of heat stress will double or triple in the coming years due to a temperature rise of 3 °C, compared to 1.5 °C. It is precisely in cities that this plays out; it is there that the phenomenon of the urban heat island shows its greatest effects [15,16]. The urban heat island is influenced by many factors; among the most relevant, we find strong soil sealing and the reduced presence and often localized green space, which generate a different energy balance than in the surrounding rural areas. Around 54% of the world's population lives in cities [17]. The percentage of the urban population is even higher in Europe (74.5%), with a value in 2018 of 70.4% in Italy (i.e., about 42 million people living in cities) [18], projected to be around 80% by 2050 [17]. As a result, cities with dense buildings and high rates of air pollution have been on the rise in recent years, and there is a risk that this trend will continue. The strong urban growth that has taken place in recent decades risks becoming a paradox of human ingenuity. Cities were built to exploit surrounding resources and to concentrate services and populations, while at the same time promoting trade and commerce. The city today has an extremely complex and difficult situation to control, upon which the impacts of human activities weigh more and more heavily. The urban heat island, together with global warming, the pollution of the ecosystems on which they depend, and increasingly frequent heat waves, risks making cities inhospitable and extremely harmful places in the coming decades [19]. The development of urban spaces, therefore, will have positive or negative effects on the quality of human life and they should be considered in the design and redevelopment of living spaces. Planning has a crucial role, especially if territorial transformations are no longer linked to mainly expansive logics. In this case, it is possible to orient urban choices and policies towards a sustainable use of resources, including the land resource that continues to be overexploited [20].

For obvious latitudinal reasons, cities in southern Europe show greater risk scenarios than those located further north, especially those whose soils are strongly sealed [21]. Italy is one of the countries with one of the highest levels of soil sealing in Europe [22], but also one of those wherein the complex morphological articulation of the territory often exposes cities to other types of risk (hydraulic risk, landslides, and earthquakes) as the latest events of 2023 (flooding in the Marche, Emilia Romagna and Tuscany) have clearly shown [23–25]. According to the 2023 report prepared by the Higher Institute for Environmental Research and Protection (ISPRA) [26], the highest percentage values of land consumption are recorded in Northern Italy. However, while in the north-west regions we are witnessing a phase of slowing growth, in the areas of Triveneto and Emilia-Romagna there is a high rate of land consumption, mainly due to the continuous urban spread that can be found in the Padano-Veneto plain. Emilia-Romagna was the fourth Italian region in terms of net land consumption in 2022, with over 635 hectares cemented in a single year, an increase of 0.35% compared to the previous year. In just a few years, it has reached 8.9% of land consumed compared to a national average of 7.14%.

Starting from these assumptions, the present work intends to analyze the relationship between the LST extracted from the data provided by MODIS and the level of soil sealing within the city of Ferrara. Many of the recent studies analyze the relationship between actual impermeable surfaces and LST. This paper focuses on potential impacts on the UHI of unbuilt but nevertheless planned impermeable surfaces.

This research was conducted over a period between 2015 and 2021, and our study aims to evaluate the potential impacts of the current urban plan's forecasts on the UHI.

The aim is to evaluate how the urban planning tool could change the current thermal pattern of the city of Ferrara. The outcomes are helpful for understanding the potential effects of new planned urban areas (which have significant implications for urban planners and decision makers), and identifying the critical territorial sectors are in need of sustainable mitigation actions.

## 2. Study Area

The study area is the city of Ferrara, capital of the province of the same name in Emilia-Romagna, Italy, which covers 405 km$^2$ (17th in terms of size at the national level). As shown in Figure 1, the territory is completely flat, with an altitude between 2.4 and 9.0 m above sea level. The city is bordered to the north by the Po River, which separates it from the Veneto region; to the south, it is bordered by the Metropolitan City of Bologna. The municipality is about 50 km from the Adriatic coast.

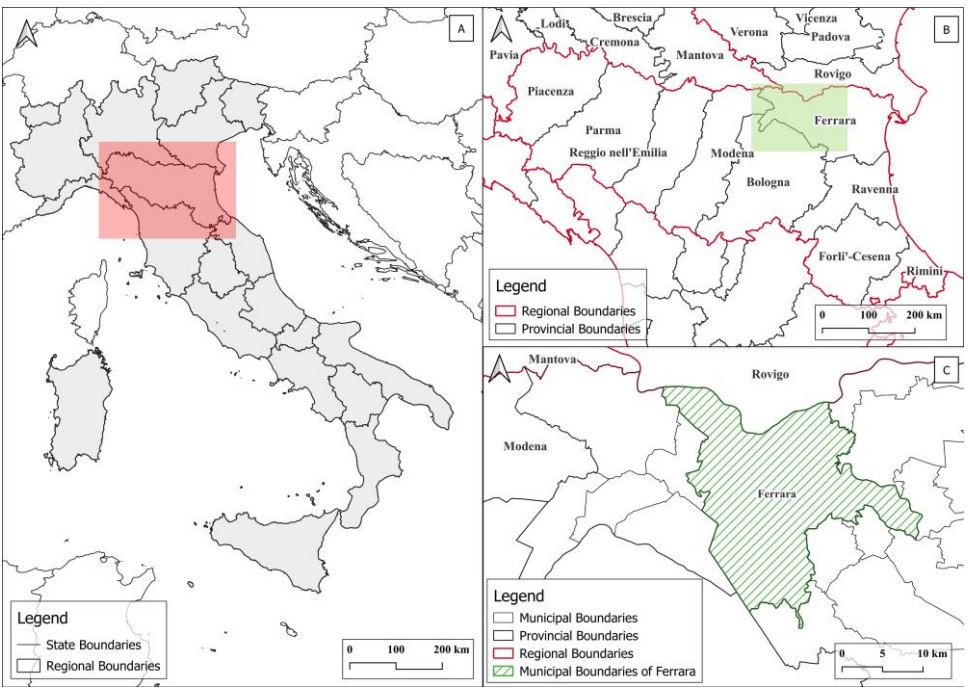

**Figure 1.** Study area: (**A**) Italian regional boundaries; (**B**) provincial boundaries; (**C**) city of Ferrara.

The current resident population is 128,990 inhabitants (ISTAT—National Institute of Statistics reference date: 31 July 2023) [27], while the population density is 318.47 inhab/km$^2$, higher than the national average value (195.41 inhab/km$^2$) and the regional one (196.78 inhab/km$^2$). The mosaic relating to land use sees a predominance of arable land (about 70%) followed by land for urban purposes (18%), while only a few lands are covered by forests (1.3%) (Source: ISPRA—Higher Institute for Environmental Research and Protection—2022) [28]. For the purpose of this work, we focus on the main green areas of Ferrara City. Figure 2 shows these areas. Green areas are well distributed in special terms. The "Giorgio Bassani" Urban Park has an extension of 1200 hectares and is located to the north, between the city walls and the Po river. The "Massari" Park, has an area of 4 hectares and represents the largest park within the city walls. There is also the "Pareschi" Park along "Corso della Giovecca", which extends for about 1.4 ha.

The city of Ferrara, according to the climatic classification introduced by the Decree of the President of the Republic no. 412/1993, is in zone E, which is characterized by 2326 Degree Day (GG). Ferrara's climate is semi-continental, with cold and wet winters and hot and muggy summers. The feeling of heat is exacerbated by the humidity of the air and poor ventilation, typical conditions of the Po Valley. Recently, the highest recorded temperature was 41.4 °C, on 2 August 2017 [29]. The average annual rainfall is around 650 mm, with a relative minimum in July and moderate peaks in October and November.

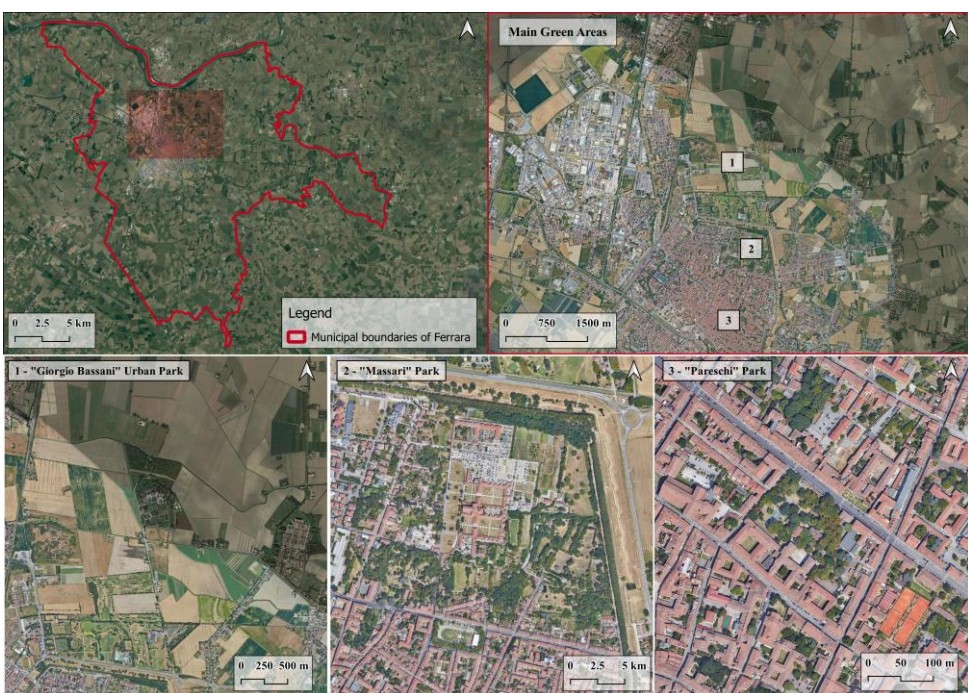

**Figure 2.** The city of Ferrara and its urban parks.

From a programmatic point of view in the municipality of Ferrara, the Municipal Structural Plan (P.S.C.) came into force in 2009. From 2009, two Municipal Operational Plans (P.O.C.) (the first expired in 2017, the second in 2022) have detailed the P.S.C. Now, the General Urban Plan (P.U.G.) is in progress as required by the new urban planning law of the Emilia-Romagna region (L.R. 24/2017).

The economic framework of the city shows that the income of the study area is historically linked to the primary sector. However, there are also industrial plants in the petrochemical sector, and a hub for small- and medium-sized enterprises. The per capita income of the Municipality is EUR 24,346 (Ministry of Economy and Finance—M.E.F. 2021 data) [30], which is higher than the national average value (EUR 19,129) and that of the Emilia-Romagna region (EUR 21,709).

## 3. Materials and Methods

### 3.1. Data on Land Consumption

The study uses data for 7 time periods (2015–2016–2017–2018–2019–2020–2021). The sealed surfaces were derived from the data made available by the Higher Institute for Environmental Research and Protection (ISPRA) [28], while the data on the LST were obtained from the NASA Earth Data portal [31]. Both datasets are open and freely downloadable. In detail, the sealed surfaces are raster data with a spatial resolution of 10 m, processed by ISPRA based on data from the National System for Environmental Protection (SNPA). The calculations relating to land consumption were carried out considering the classes shown in Table 1:

**Table 1.** Legend of ISPRA land consumption for the study area.

| Code | Description |
| --- | --- |
| 1 | Consumed soil |
| 11 | Permanent Consumed Soil |
| 111 | Buildings, Warehouses |
| 112 | Paved roads |

**Table 1.** *Cont.*

| Code | Description |
|------|-------------|
| 113 | Railway headquarters |
| 114 | Airports |
| 115 | Harbours |
| 116 | Other sealed areas |
| 117 | Paved permanent greenhouses |
| 118 | Landfills |
| 122 | Yards |
| 125 | Ground-mounted photovoltaic fields |

### 3.2. Data on Land Temperature

MODIS images have a geometric resolution of 1 km/pixel and have a temporal resolution of 8 days. These images provide LST (degrees kelvin) values for both day and night. LST is a measure of the amount of heat on the Earth's surface. It is a physical quantity that reflects the thermal energy emitted or absorbed by the Earth's surface. LST is influenced by various factors, including solar irradiance, light reflection, atmospheric transmission, and the thermal properties of the soil.

Its measurement is essential for understanding climate change, monitoring the seasons, studying meteorological phenomena, and assessing environmental impacts [32,33]. According to an analysis conducted on historical temperature data by the website 3Bmeteo [34], It was found that the warmest temperature of the year was recorded between the last week of July and the first week of August.

### 3.3. Comparison between MODIS Data and Land Consumption

Based on this information, the relevant MODIS data for this period were analyzed by arithmetic averaging the daytime and night-time temperature values. For the comparison between the LST data and the land consumption in the 7 time-sections, the territory of the city of Ferrara was divided into 1 km side squares (for a total of 566 cells), corresponding to the resolution of the MODIS data and the geography of the cells that make up the image. To avoid small portions of territory, corresponding to the intersection between the 1 km$^2$ grid and the municipal boundaries, cells outside the municipal boundary were also considered (Figure 3).

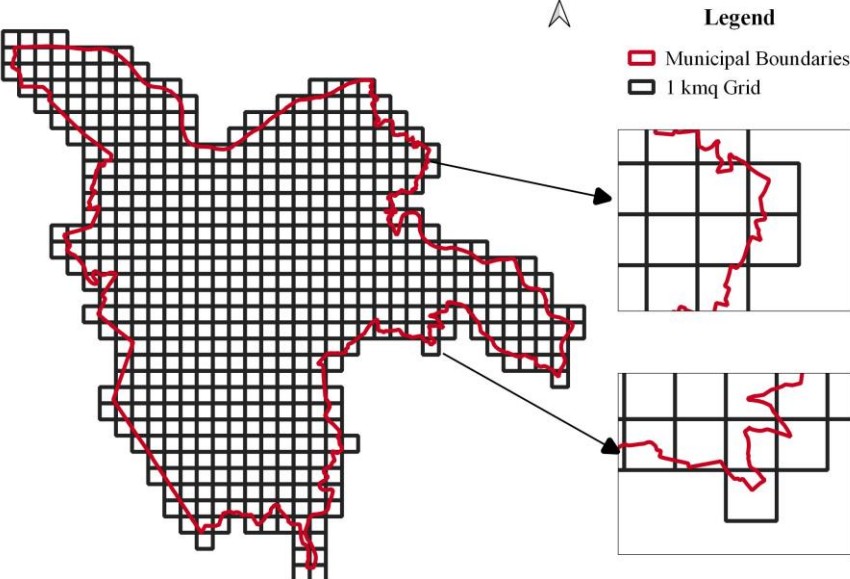

**Figure 3.** Subdivision of the municipality of Ferrara into a grid of 1 km$^2$.

To each grid's cell, using QGIS software (Version 3.22 "Hannover"), the following data were associated: average LST value for each year derived from the data of the indicated week (in degrees Celsius), and coverage of sealed surfaces (in $m^2$) derived from the layer relating to land consumption already mentioned above. The final database contains the data just described for each year examined. A sealed surface is defined as "the surface layer of the soil, removed and replaced with impermeable material". This type of surface therefore has a direct impact on the thermal comfort of an area due to a poor ability to reflect heat and high thermal conductivity that favors the release of heat during night-time hours.

### 3.4. Data on Municipality's Urban Planning

The municipality's urban planning tool was used to identify the possible future settlement structure derived from the urban planning forecasts currently in force. Among the synoptic items reported in this instrument, the one indicated "Urbanizable land" basically indicates which portions of land can be transformed for urban uses. In addition, a comparison with land use cartography (produced by ISPRA with a resolution of 10 m/pixel) has made it possible to identify which types of land use could be most affected by the implementation of urban planning forecasts. The considered land use categories are shown in Table 2.

**Table 2.** Legend of ISPRA land uses for the study area.

| Code | Description |
|---|---|
| 2 | Forest Use |
| 3 | Quarries and mines |
| 4 | Urban and related areas |
| 5 | Water Uses |
| 11 | Arable crops |
| 12 | Fodder |
| 13 | Prevailing crops |
| 14 | Agroforestry Areas |
| 16 | Other Agricultural Uses |
| 61 | Wetlands |
| 62 | Other non-economic uses |

### 3.5. Sealing Density Index

To understand both the type of relationship and the degree of correlation between sealed surfaces and temperature, the value of the soil sealing density index (Formula (1)) for each year between 2015 and 2021 was calculated for each cell. The sealing density [26] is defined as follows:

$$D_{imp} = \frac{S_{imp\_x}}{S_{grid}} [\%] \tag{1}$$

$S_{imp\_x}$ = sealed surface at x year [ha];
$S_{grid}$ = cell's grid surface [ha].
The index values were grouped into the following 7 macro-classes:

- Class 1: $D_{imp} \geq 70\%$;
- Class 2: $50\% \leq D_{imp} < 70\%$;
- Class 3: $35\% \leq D_{imp} < 50\%$;
- Class 4: $20\% \leq D_{imp} < 35\%$;
- Class 5: $10\% \leq D_{imp} < 20\%$;
- Class 6: $5\% \leq D_{imp} < 10\%$;
- Class 7: $D_{imp} < 5\%$.

The sealing density value decreases from class 1 (>70%) to class 7 (<5%). The lower the level of sealing, the greater the coverage of the other types of soil cover. Figure 4 shows the spatial distribution of $D_{imp}$ value. The approach adopted made it possible to construct

a table with the two considered variables, and to analyze the possible correlations among them. The correlation was evaluated through the coefficient of determination ($R^2$), i.e., the index that measures the link between the variability of the data and the correctness of the statistical model used. If the value of $R^2$ is close to 1, it means that the regressors predict well the value of the dependent variable in the sample; if the value is 0, the above condition does not occur.

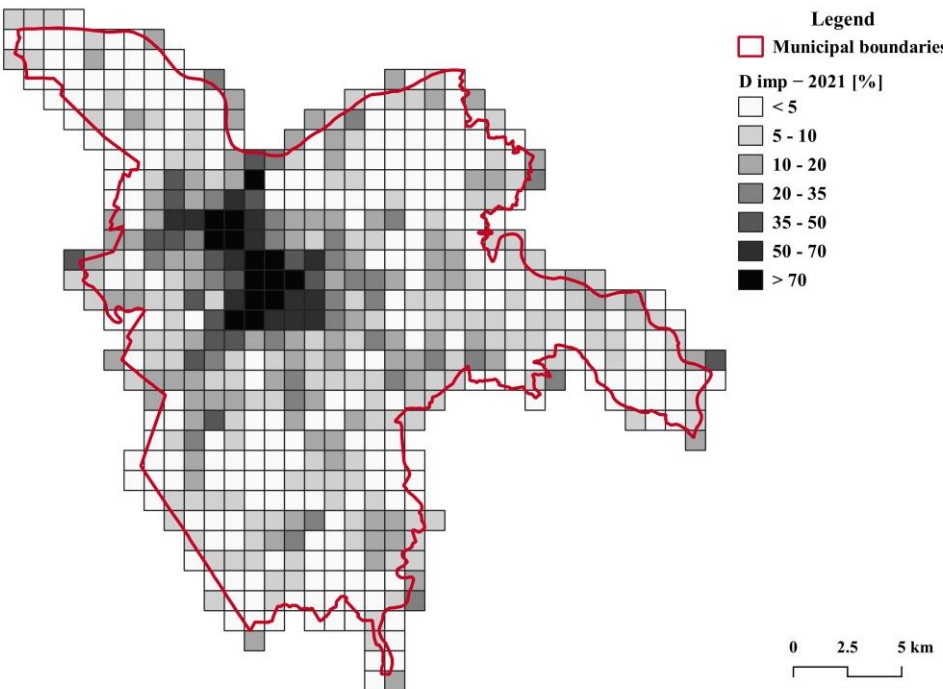

**Figure 4.** Spatial distribution of sealing density at 2021.

## 4. Results

Between 2015 and 2021, the city of Ferrara recorded an increase of about 43 ha in sealed surfaces, with an almost constant increase over the 7 years analyzed. The city's sealing density increased from 13.5% in 2015 to 13.6% in 2021, against a decrease in the population of almost 4000 inhabitants. For urban areas of this size, this kind of demographic dynamics are typical, and specifically, it has been detected an increase in the number of residents in the territories of the first urban belt (lower costs related to housing) consequently generates commuting phenomena, and indirectly, land consumption.

Table 3 shows the average temperature values recorded in the 7 years analyzed according to the previous classification.

**Table 3.** Average temperature values recorded in the 7 years in Celsius degree.

|   |   | 2015 | 2016 | 2017 | 2018 | 2019 | 2020 | 2021 |
|---|---|------|------|------|------|------|------|------|
| 1 | $D_{imp} > 70$ | 30.14 | 30.35 | 33.03 | 32.32 | 29.13 | 30.85 | 30.33 |
| 2 | $50 < D_{imp} < 70$ | 29.60 | 29.87 | 32.41 | 31.76 | 28.62 | 30.19 | 29.43 |
| 3 | $35 < D_{imp} < 50$ | 28.22 | 28.57 | 31.35 | 30.60 | 27.63 | 28.74 | 28.69 |
| 4 | $20 < D_{imp} < 35$ | 27.67 | 28.08 | 30.61 | 29.95 | 27.15 | 28.30 | 28.17 |
| 5 | $10 < D_{imp} < 20$ | 27.18 | 27.76 | 30.23 | 29.62 | 26.80 | 27.90 | 27.90 |
| 6 | $5 < D_{imp} < 10$ | 27.06 | 27.59 | 29.96 | 29.29 | 26.70 | 27.66 | 27.75 |
| 7 | $D_{imp} < 5$ | 26.91 | 27.55 | 29.89 | 29.33 | 26.69 | 27.71 | 27.69 |

It is possible to deduce the increasing average temperature values recorded as the level of soil sealing increased from the table. There is a thermal gradient with thermal differences of the order of degrees, depending on the greater or lesser presence of sealed surfaces. Over the time frame studied, we observed a variation in average temperature among grid cells with $D_{imp} > 70\%$ and $D_{imp} < 5\%$ of 2.91 °C. This result is clear by correlating the quantities analyzed, as shown in the graph in Figure 5.

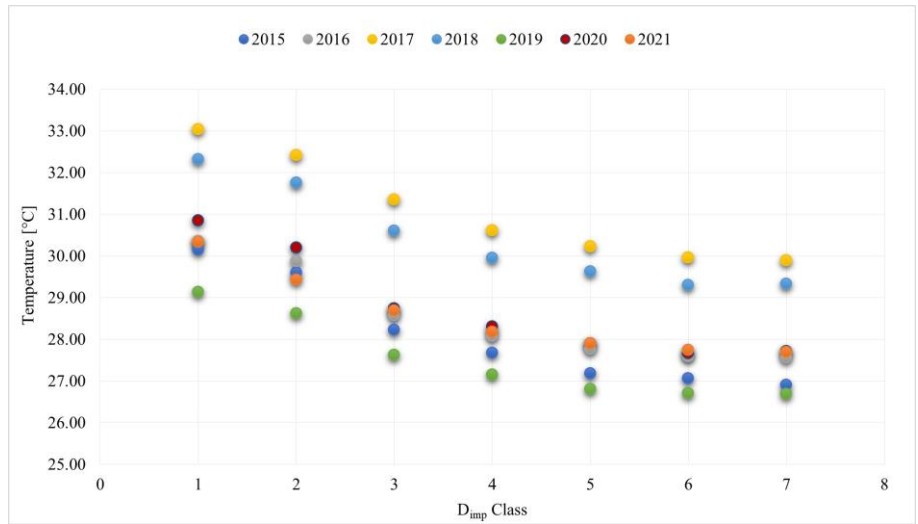

**Figure 5.** Comparison between soil sealing and average temperature in grid's cells.

Table 4 shows the values of the correlation coefficient ($R^2$) for each analyzed year.

**Table 4.** Correlation coefficient ($R^2$) between urbanization level and temperature.

| Year | $R^2$ |
| --- | --- |
| 2015 | 0.8975 |
| 2016 | 0.8746 |
| 2017 | 0.9198 |
| 2018 | 0.9038 |
| 2019 | 0.8799 |
| 2020 | 0.8667 |
| 2021 | 0.8808 |

For all considered ranges, the value of $R^2$ is always greater than 0.86. A maximum correlation value of almost 0.92 was recorded for 2017, the year in which the highest average temperature was recorded. The distribution of frequencies, for both variables investigated, shows a non-uniform trend. For this reason, the choice of the interval values was made using the natural breaks algorithm (also known as Jenks' optimization); it is based on natural groupings inherent in the data. Class breaks that best group's similar values and maximizes the differences between classes. In this way, 3 sealing density ranges and 3 average surface temperature ranges for the year 2021 were defined. Table 5 shows the considered ranges (reference year, 2021), while Figure 6 shows the geographical spatialization of the analysis carried out.

**Table 5.** Range of temperature and $D_{imp}$ for 2021, used for the map in Figure 6.

| T Range [°C] | $D_{imp}$ Range [%] |
| --- | --- |
| 25.0–27.6 | <20 |
| 27.6–28.8 | 20–50 |
| 28.8–30.9 | >50 |

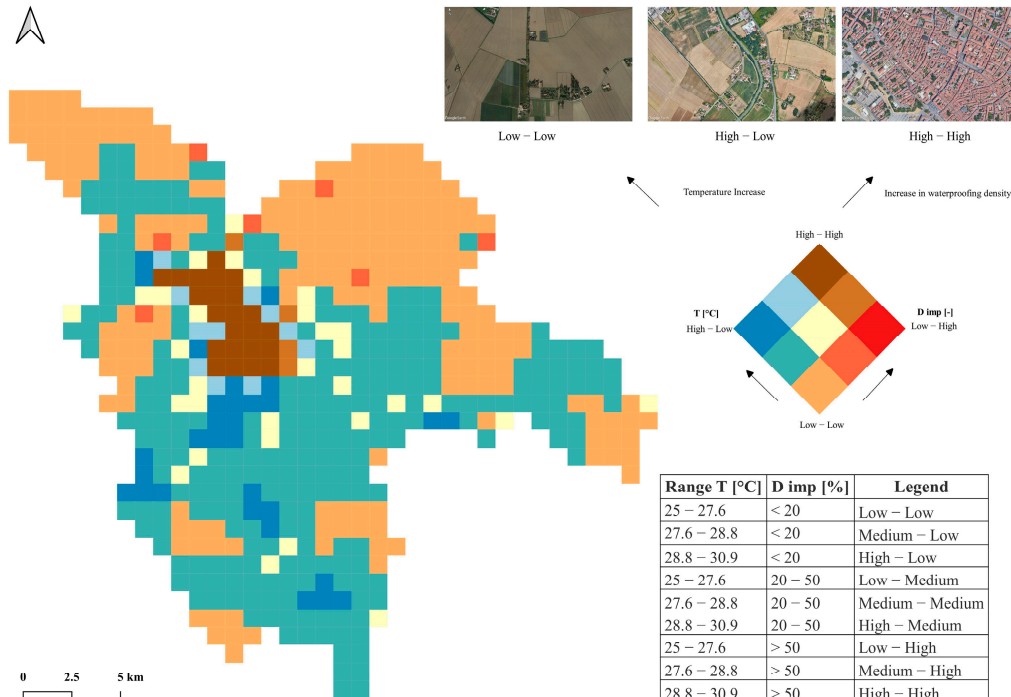

**Figure 6.** Spatial correlation between sealing density and predefined temperature ranges.

It is evident that the high–high combination characterizes the entire urban area of Ferrara. It is here that the highest temperatures and greater soil sealing are recorded. The areas in the northeast, on the other hand, are those where the low–low combination is present with greater continuity. It is an area characterized by a strong arable connotation, where the settlement manifests itself in a small form and disperses in the forms of individual buildings or some aggregates. The eco-mosaic that characterizes the high–low condition located in the south-eastern part of the main urban area shows a settlement that is more compact (there are several hamlets in these territories) and fades towards the dispersed and poorly organized typology typical of sprinkling [35]. The high temperatures recorded in these areas are affected by the adjacent main urban area, while the green areas are few, and their mitigation action is scarce.

Figure 6 shows that there are no "low–high" cells, i.e., cells characterized by a high sealing density (>50%) and values lower than the temperature range (25–27.6 °C). As the analyses show, there is a strong dependence between the level of soil sealing and the LST. The configuration of the settlement also plays an important role in this regard. The settlement development forecasts present in the urban planning tool, evaluated not only in terms of quantities but also in terms of the methods of implementation on the territory, could strongly affect the future climatic structure of the area. Figure 7 shows the sealing surface provided for by the current urban planning instrument (in red) compared to that recorded in 2021 (in black). The full implementation of the P.S.C. could lead to an increase in the sealed area of about 1335 ha, increasing the $D_{imp}$ from the current 13.59% to 17.05% (an increase of 3.5 percentage points). The most widely used land use class for such transformations is arable land (90.46%). A smaller percentage is occupied by forage (4.02%), and the remaining 5.52% by the

others land use classes. The number of cells classified as above changes, as shown in Table 6. It should be noted that the number of cells where the $D_{imp}$ value is higher than 70% would double, with obvious repercussions for the climate of the areas involved. Figure 8 shows the spatial distribution of the cells with a $D_{imp} > 50\%$, according to the full implementation of the P.S.C.

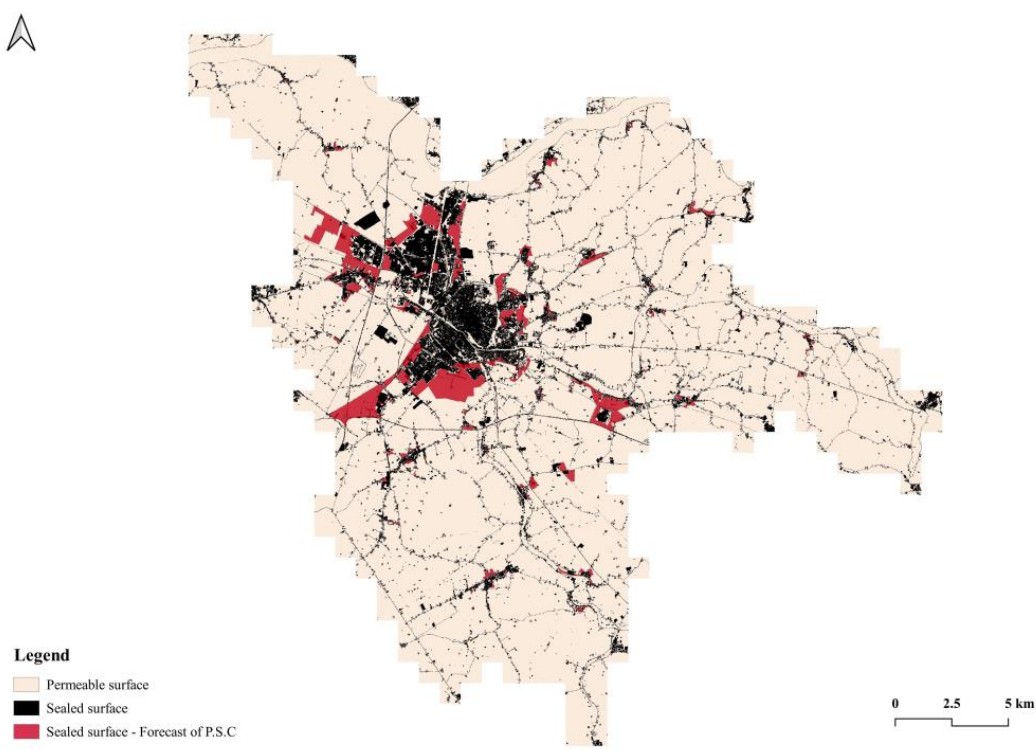

**Figure 7.** Geography of sealed areas in 2021 (in black) and urban forecast of P.S.C. (in red).

**Table 6.** Number of grid's cells for each class of $D_{imp}$ measured for 2021, and with the full implementation of P.S.C.

|  | N° Cells—2021 | N° Cells—P.S.C. |
|---|---|---|
| $D_{imp} > 70$ | 14 | 32 |
| $50 < D_{imp} < 70$ | 13 | 13 |
| $35 < D_{imp} < 50$ | 18 | 20 |
| $20 < D_{imp} < 35$ | 31 | 43 |
| $10 < D_{imp} < 20$ | 90 | 82 |
| $5 < D_{imp} < 10$ | 148 | 127 |
| $D_{imp} < 5$ | 252 | 249 |

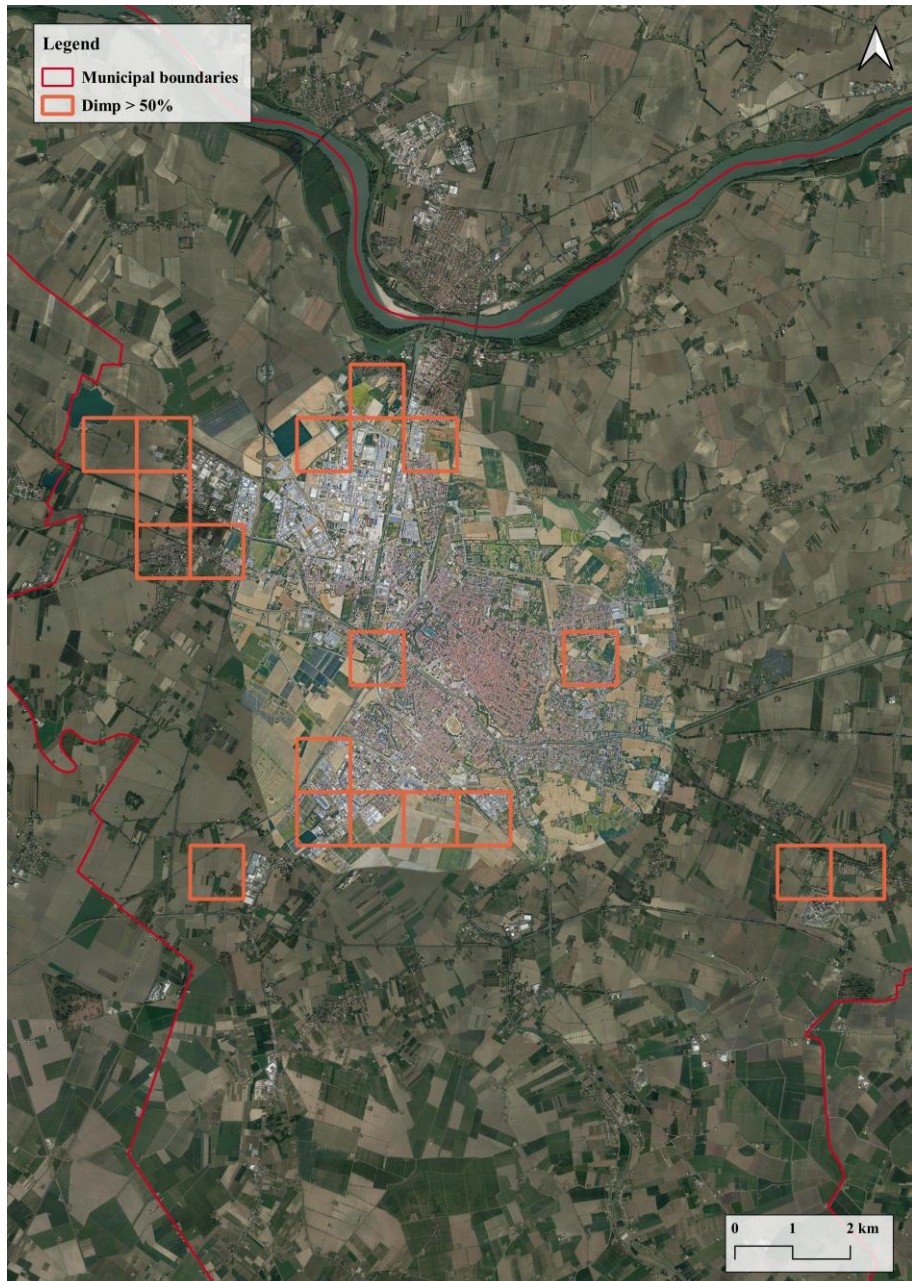

**Figure 8.** Spatial distribution of $D_{imp} > 50\%$ (full implementation of P.S.C.).

## 5. Discussion

This work highlighted the strong dependence that exists between soil sealing and the high temperatures that are found today in global urban contexts [36–38]. The focus of the study was on the effects that planning choices could have on the thermal comfort of the municipality of Ferrara.

The effects of sealed areas on UHI have been examined in others works. However, our work also studies the possible impact of scenarios linked to urban planning forecasts. The authors of [39], using MODIS data, identified the effect of the spatial grouping of urban surfaces on the LST, for different climate contexts, in seven US metropolitan cities. According to this study, dispersed sealed surfaces diminish the effect of LST. However, this study does not consider the effect that such dispersion may subsequently have on soil consumption, related ecosystem services, and environmental fragmentation. The authors of [40] estimate the effect of urban parks on UHI, compared to the LST. This study,

using data measured by on-site sampling, demonstrates how urban parks provide cooling benefits up to about 800 m from their boundary. It also suggests that, for a mitigation of the effect of UHI, in urban planning strategies, smaller but more dispersed parks are more preferable than single, but larger parks. The authors of [18] introduce the tree cover density index to study the effect of urbanization on the UHI phenomenon in ten Italian cities. The authors show that the presence of sealed surfaces and low tree density are the main processes that increase the intensity of UHI. This study, however, only considers these two factors, excluding other elements of urban configuration. Table 6 shows that in a hypothetical scenario with a strong increase in cells with a sealing density value higher than 50%, cells showed the highest temperature values recorded within the entire period investigated. We do not have implementation plans that would allow a more detailed analysis, but in any case, in these areas, the same conditions already found and discussed in the work could occur. For the above reasons, the analysis carried out on soil sealing was only quantitative and not configural. However, clearly, the construction methods of the interventions, the building types and the management of the appurtenant spaces also play an important role in influencing the thermal structure of the area [41,42]. The data on temperature come from MODIS, which has a spatial resolution of 1 km × 1 km. Today, data with a higher resolution are available. For example, there are Landsat data with a spatial resolution of 30 m. The use of this kind of data certainly produces more accurate results, particularly in the identification of urban areas in which the urban heat island occurs. However, on the other hand, the objective of this work is to study the municipality of Ferrara from a planning scale, and the use of MODIS data, which has a lower resolution, is sufficient to show the phenomenon at this scale. The use of higher resolution data did not justify the computational burden that its use would have entailed.

As can be seen from Figure 7 (surfaces in red), the areas of new urban forecast are in continuity with the pre-existing urban areas. If, on the one hand, this method of construction decreases land consumption and urban sprawl, on the other hand, it could generate changes in the effects of UHI even in areas that today find themselves in different thermal conditions. The increase in impermeable surfaces in the main urban area would probably extend the high–high condition to a larger portion of the territory, thus exposing the population living in this area to a higher risk. If the demographic dynamics from ISTAT data in the last decade are analyzed, it can be seen that the population has decreased by more than 3000 units in the municipality. However, considering that the vast municipal territory consists of about 40 inhabited localities, these data deserve a more careful analysis. In particular, the main urban area recorded a sharp decrease in residents (over 1400 fewer today than in 2013) for reasons certainly related to the livability of the area, while some hamlets not far from the city center (e.g.,: Fossanova San Marco and San Martino) recorded an increase in the number of residents. The two reported areas are in a different environmental context, predominantly agricultural with a good presence of green spaces and trees in the urban context (Figure 9).

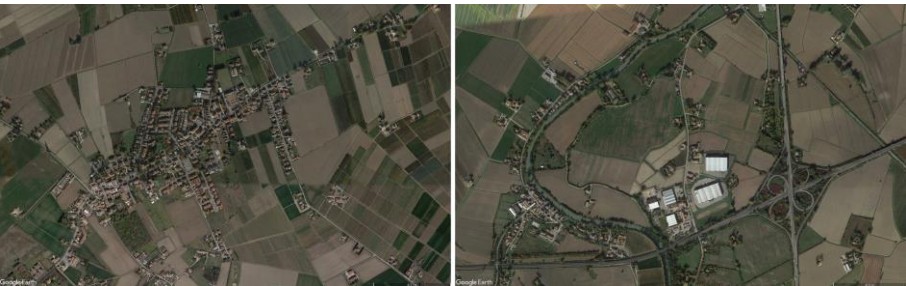

**Figure 9.** The hamlet of San Martino (on the **right**) and Fossanova San Marco (on the **left**).

## 6. Conclusions

The used methodology can be replicated in other contexts, and it can provide interesting information about the thermal planning of a territory. The amount of sealed surface

and its geographical distribution have a leading role in influencing UHI (see Table 4), even if these are not the only factors to influence LST variations in urban areas. For this reason, the creation and the analysis of urban development models based on urban forecasts is fundamental. This kind of models represent a useful tool to identify effective strategies of contrast/adaptation to the effects of UHI. The possibility to identify in a preventive way, which areas could undergo changes in their current thermal pattern allows the municipality to intervene through specific indications. Specifically, municipalities can implement plans considering these factors with the aim to mitigate UHI impacts. As already indicated in the work, the urban planning tool used does not allow us to trace the future urban layout of Ferrara exactly, but it does allow us to understand which areas will be involved in these developments. The results of the work clearly suggest the careful planning of the new parts of the city, which will have to be designed through nature-based solutions (NBS), with large green spaces and permeable areas. The aim of organizing the territory in this way is to not increase the impermeable surfaces that could strongly extend the critical areas identified by the model as high–high, with important consequences for the city's thermal structure. The proposed methods represent a useful decision support system. Specifically, these methods can be enhanced, considering other variables strictly linked to UHI; in this way, the potential effects of new planned urban areas can be explored in advance.

**Author Contributions:** Conceptualization, G.D.P. and F.Z.; methodology, G.D.P. and F.Z.; software, G.D.P.; validation, F.Z. and C.M.; formal analysis, G.D.P. and E.M.; investigation, G.D.P. and C.M.; data curation, G.D.P. and E.M.; writing—original draft preparation, G.D.P., E.M. and F.Z.; writing—review and editing, C.M.; visualization, G.D.P. and C.M.; supervision, F.Z. All authors have read and agreed to the published version of the manuscript.

**Funding:** This research received no external funding.

**Institutional Review Board Statement:** Not applicable.

**Informed Consent Statement:** Not applicable.

**Data Availability Statement:** The data presented in this study are available on request from the corresponding author.

**Conflicts of Interest:** The authors declare no conflict of interest.

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
