# Peer review of "Land Surface Temperature and Urban Policies: The Ferrara City Case Study"

_sustainability, doi:10.3390/su152416825_

Round 1

Reviewer 1 Report

Comments and Suggestions for Authors

I would like to thank the authors for considering “Sustainability” for publishing their paper. I found the idea clear, interesting, and practical. I added my comments here to better prepare the article for potential publication:

Line 35 to 38: Are there any relevant references here to support the claim?

Line 45: So, is the heat island due to soil sealing, or is it one of the contributing factors?

Line 47: LST stands for land surface temperature, which includes different land covers rather than only soil surface temperature. Please consider changing the terms.

Line 56: Replace "build" with "Built." Please proofread the entire document for consistency.

Line 68 to 70: Are there any references to support these claims?

Line 84: Please see the comment above. Also, use only the abbreviation (LST).

Lines 83 to 89: The objectives of the study are not well defined; they are vague and lengthy. Please include the following:

a) Address the research gap not covered by previous studies (a concise two-three lines).

b) Define one or two specific objectives. For example, the study aims to define... to address or evaluate... by...

c) Clearly state the direct and potential contributions of the study. For example, the results provide insights for policymakers to... and can potentially be incorporated as a... to highlight anthropogenic footprints...

Line 98: Include a, b, c for your figures and then explain which is which.

Line 133: Figure 2 quality is low. Please use around 500dpi quality images.

Line 137: Again, another point of confusion. Is it the surface temperature of the soil, land surface temperature, or soil surface temperature? Please be consistent.

Line 145: I am not sure why MODIS data were used. With the availability of Landsat data, where LST can be easily calculated at a 30-meter resolution, using 1km images seems illogical.

Line 251: Please enlarge the table and use a better layout for clarification. Also, what happened to D imp > 70 percent in your Table?

Line 319: Almost all figures are low quality. Please consider using bigger and better quality figures

Author Response

Reviewer 1

I would like to thank the authors for considering “Sustainability” for publishing their paper. I found the idea clear, interesting, and practical. I added my comments here to better prepare the article for potential publication:

Reply: Many thanks for your valuable comments. They have significantly ameliorated our research paper.

Line 35 to 38: Are there any relevant references here to support the claim?

Reply: Thank you for your comment, we add different references for this part of the paper [8-11].

Pappalardo, S.E.; Zanetti, C.; Tedeschi, V. Mapping urban heat islands and heat-related risk during heat waves from a climate justice perspective: A case study in the municipality of Padua (Italy) for inclusive adaptation policies. Land Urb. Plan. 2023, 238. https://doi.org/10.1016/j.landurbplan.2023.104831.

Kornhuber, K.; Lesk, C.; Schleussner, C.F.; Jägermeyr, J.; Pfleiderer, P.; Horton, R.M. Risks of synchronized low yields are underestimated in climate and crop model projections. Nat. Comm. 2023, 14(1). https://doi.org/10.1038/s41467-023-38906-7.

Niggli, L.; Huggel, C.; Muccione, V.; Neukom, R.; Salzmann, N. Towards improved understanding of cascading and interconnected risks from concurrent weather extremes: Analysis of historical heat and drought extreme events. PLOS Clim 2022, 1(8). https://doi.org/10.1371/journal.pclm.0000057

Laino, E.; Iglesias, G. Extreme climate change hazards and impacts on European coastal cities: A review. Ren. Sust. En. Rev. 2023, 184. https://doi.org/10.1016/j.rser.2023.113587

Line 45: So, is the heat island due to soil sealing, or is it one of the contributing factors?

Reply: thank you for your comment, we detailed that these are some of the factors that affect the urban heat island.

“The urban heat island is influenced by many factors , among the most relevant we find the strong soil sealing, the reduced presence and often localized green space that generates a different energy balance than in the surrounding rural areas”.

Line 47: LST stands for land surface temperature, which includes different land covers rather than only soil surface temperature. Please consider changing the terms.

Reply: Thank you for your comment, we have changed the term as suggested

Line 56: Replace "build" with "Built." Please proofread the entire document for consistency.

Reply: thank you for your comment. we have changed the term as suggested.

Line 68 to 70: Are there any references to support these claims?

Reply: Thank you for your comment, we add 2 references for this part of the paper [22-23].

Ferreira, C.S.S.; Seifollahi-Aghmiuni, S.; Destouni, G.; Ghajarnia, N.; Kalantari, Z.  Soil degradation in the European Mediterranean region: Processes, status and consequences. Sc. Tot. Env. 2022, 805. https://doi.org/10.1016/j.scitotenv.2021.150106

European Environment Agency (EEA). Avaible online: https://www.eea.europa.eu/data-and-maps/daviz/percentage-sealing-by-country-1#tab-chart_5 (accessed on 19 October 2023).

Line 84: Please see the comment above. Also, use only the abbreviation (LST).

Reply: Thanks for your comment. We have used only the abbreviation.

Lines 83 to 89: The objectives of the study are not well defined; they are vague and lengthy. Please include the following:

  1. a) Address the research gap not covered by previous studies (a concise two-three lines).
  2. b) Define one or two specific objectives. For example, the study aims to define... to address or evaluate... by...
  3. c) Clearly state the direct and potential contributions of the study. For example, the results provide insights for policymakers to... and can potentially be incorporated as a... to highlight anthropogenic footprints...

Reply: Thanks for your comment. We have added a short paragraph (line 89 to 100)

Many of the recent studies analyze the relationship between actual impermeable surfaces and LST. This paper focuses on the potential impacts on the UHI of unbuilt but nevertheless planned impermeable surfaces. This research was conducted over a period between 2015 and 2021 and our study aims to evaluate the potential impacts of the current urban plan’s forecasts on the UHI. The aim is to evaluate how the urban planning tool could change the current thermal asset pattern of the city of Ferrara. The outcomes are helpful for understanding the potential effects of new planned urban areas which have significant implications for urban planners and decision-makers and identifying the critical territorial sectors in need of sustainable mitigation actions.  

Line 98: Include a, b, c for your figures and then explain which is which.

Reply: Thank you for these suggestions, we make the changes you request.

Line 133: Figure 2 quality is low. Please use around 500dpi quality images.

Reply: thank you, the images are already at 500dpi, maybe the problem is the .pdf format.

Line 137: Again, another point of confusion. Is it the surface temperature of the soil, land surface temperature, or soil surface temperature? Please be consistent.

Reply: Thanks for your precious comment. We have used the land surface temperature. We have updated our text.

Line 145: I am not sure why MODIS data were used. With the availability of Landsat data, where LST can be easily calculated at a 30-meter resolution, using 1km images seems illogical.

Reply: thank you for your precious comment. We add a short paragraph in the article to justified our choice. The paragraph is the follow.

“The data on temperature come from MODIS, that has a spatial resolution of 1kmx1km. Today a data with an higher resolution are available. For example, there is Landsat data with a spatial resolution of 30 meters. The use of this kind of data certainly produces a more accurate results, particularly in the identification of urban areas in which the urban heat island occurs. But on the other hand, the objective of this work is to study the municipality of Ferrara from a planning scale, and the use of MODIS data, that has a lower resolution, is sufficient to show the phenomenon at this scale. The use of higher resolution data did not justify the computational burden that its use would have entailed.”

Line 251: Please enlarge the table and use a better layout for clarification. Also, what happened to D imp > 70 percent in your Table?

Reply: thank you for your comment. The layout of the table has been changed, the table has been enlarged. For the graphical representation has been used the range in table 5. The choice of a different classification is due to the better understanding of the results.

Line 319: Almost all figures are low quality. Please consider using bigger and better quality figures

Reply: thank you, the images are already at 500dpi, maybe the problem is the .pdf format.

Reviewer 2 Report

Comments and Suggestions for Authors

The objective of this study is to examine the correlation between LST derived from MODIS data, and the extent of soil sealing in the municipality of Ferrara. While the authors have undertaken a comprehensive investigation, I still harbor some concerns regarding the manuscript.

 L30-33: To elucidate the context of global urban expansion trends and challenges, the following papers are recommended:

"Evaluating trends, benefits, and risks of global cities in recent urban expansion for advancing sustainable development" in Habitat International…

"Mapping global urban land for the 21st century with data-driven simulations and Shared Socioeconomic Pathways" in Nat. Commun…

The introduction requires clarification, and the logic needs further refinement.

L47-49, L111-112, L234: Utilizing just one sentence for a graph seems awkward.

Consider merging Figure 1 and Figure 2. Using two images to introduce the study area might be unnecessary. Additionally, introducing subsections in Section 2 could enhance the clarity of the flowchart.

Please provide information on how the land use classes in Table 2 were determined.

Include assumptions or references for Eq.(1) in the methodology section.

In Section 2, introduce the method used to analyze the relationship between sealing density and temperature.

Revise the discussion section to present critical analyses of the results, critiques, and recommendations derived from the study. Compare findings with similar studies and explore broader implications for mitigating the Urban Heat Island (UHI) effect in urban areas.

Comments on the Quality of English Language

Extensive editing of English language required.

Author Response

Reviewer 2

The objective of this study is to examine the correlation between LST derived from MODIS data, and the extent of soil sealing in the municipality of Ferrara. While the authors have undertaken a comprehensive investigation, I still harbor some concerns regarding the manuscript.

L30-33: To elucidate the context of global urban expansion trends and challenges, the following papers are recommended:

"Evaluating trends, benefits, and risks of global cities in recent urban expansion for advancing sustainable development" in Habitat International…

"Mapping global urban land for the 21st century with data-driven simulations and Shared Socioeconomic Pathways" in Nat. Commun

Reply: thanks for your suggestion, clarifications have been added regarding the context of global urban expansion trends and challenges. Please, see below:

Urban sprawl is one of the most relevant in this regard as it plays a crucial role in economic development and social welfare, nevertheless, concerns about its detrimental effects on the environment, both locally and globally, are continuously increasing. The process of urban sprawl is one of the most visible, irreversible and rapid types of land cover/land use change in contemporary human history and is a key factor in many large-scale environmental and social changes. These changes often result in high temperatures, changes in precipitation, and increased concentrations of air pollution, as impervious surfaces can alter surface albedo and evapotranspiration while also releasing aerosols and anthropogenic heat.

The introduction requires clarification, and the logic needs further refinement.

Reply: Thank you for your comment. We made the speech more fluid and refined the parts.

L47-49, L111-112, L234: Utilizing just one sentence for a graph seems awkward.

Reply: Thank you for your comments.

For line: 47- 49 we change this part: “The urban heat island is influenced by many factors, among the most relevant we find the strong soil sealing, the reduced presence and often localized green space that generates a different energy balance than in the surrounding rural areas”.

For line: 111 – 112 we delete this part, because it is not useful to the comprehension of the paper.

For line: 234 we use only one sentence to describe this table because the explanation is in the materials and methods paragraph.

Consider merging Figure 1 and Figure 2. Using two images to introduce the study area might be unnecessary. Additionally, introducing subsections in Section 2 could enhance the clarity of the flowchart.

Reply: Thank you for your important comments, for a better comprehension of the images we prefer to keep the two images separate. Instead, we add subsections to materials and methods paragraph.

Please provide information on how the land use classes in Table 2 were determined.

Reply: Thank you, the land use classes in Table 2 were determined accordly to ISPRA classification available at this link: ISPRA. https://www.isprambiente.gov.it/it/banche-dati/banche-dati-folder/suolo-e-territorio/uso-del-suolo

Include assumptions or references for Eq.(1) in the methodology section.

Reply: thank you for your comment. We add the reference.

In Section 2, introduce the method used to analyze the relationship between sealing density and temperature.

Reply: thank you for your comment. The used method to analyze the relationship between sealing density and temperature, specifically, it has been explained in the subparagraph “Comparison between MODIS data and land consumption”. The choice of the 7 macro classes of sealing density is expert based, and it is linked to the better graphic comprehension by the reader of the correlation between the two variables.

Revise the discussion section to present critical analyses of the results, critiques, and recommendations derived from the study. Compare findings with similar studies and explore broader implications for mitigating the Urban Heat Island (UHI) effect in urban areas.

Reply: thank you for the suggestions. We have revised the discussion section, and we separated this section from conclusions (as indicated also by reviewer 3). We have analyzed various similar studies and we have added some considerations about them.

Reviewer 3 Report

Comments and Suggestions for Authors

The paper is overall well organized and written. I only suggest some changes and improvements regarding results and discussion section (it is easier if they appear together and if discussion is presented right after results), and conclusions section that mut be improved and presented separately from the discussion section. Some more conclusions are needed regarding the policies that could be implemented. A sound sentence conclusion should be presented at the end of the paper regarding the goal of the paper presented both in abstract and introduction section.

Author Response

Reviewer 3

Comment line 248

Reply: thanks for the comment, the result is inherent in table 3. We changed "di" to "of". It was a translation error.

The paper is overall well organized and written. I only suggest some changes and improvements regarding results and discussion section (it is easier if they appear together and if discussion is presented right after results), and conclusions section that mut be improved and presented separately from the discussion section. Some more conclusions are needed regarding the policies that could be implemented. A sound sentence conclusion should be presented at the end of the paper regarding the goal of the paper presented both in abstract and introduction section.

Reply: Many thanks for your precious comments. We have separated the discussion section from the conclusions, and we have improved both.

Round 2

Reviewer 2 Report

Comments and Suggestions for Authors

The paper is well revised.

Comments on the Quality of English Language

Minor editing of English language required.

Reviewer 3 Report

Comments and Suggestions for Authors

The authors have significantly improved the manuscript regarding all the comments provided. It is now ready to be published.